# MEMBERSHIP INFERENCE ATTACKS FOR UNSEEN CLASSES

## ABSTRACT

The state-of-the-art for membership inference attacks on machine learning models is a class of attacks based on *shadow models* that mimic the behavior of the target model on subsets of held-out nonmember data. However, we find that this class of attacks is fundamentally limited because of a key assumption—that the shadow models can replicate the target model's behavior on the distribution of interest. As a result, we show that attacks relying on shadow models can fail catastrophically on critical AI safety applications where data access is restricted due to legal, ethical, or logistical constraints, so that the shadow models have no reasonable signal on the query examples. Although this problem seems intractable within the shadow model paradigm, we find that *quantile regression* attacks are a promising approach in this setting, as these models learn features of member examples that can generalize to unseen classes. We demonstrate this both empirically and theoretically, showing that quantile regression attacks achieve up to **11× the TPR** of shadow model-based approaches in practice, and providing a theoretical model that outlines the generalization properties required for this approach to succeed. Our work identifies an important failure mode in existing MIAs and provides a cautionary tale for practitioners that aim to directly use existing tools for real-world applications of AI safety.

## 1 INTRODUCTION

Membership inference attacks (MIAs) are useful tools to predict whether or not a particular example was used when training a machine learning model (Shokri et al., 2017; Carlini et al., 2022). They are commonly used to evaluate various forms of AI safety risk, such as the strength at which a model preserves privacy or the likelihood that a model was trained on certain unsafe data. MIAs take advantage of the fact that models tend to memorize training data, so that even when a model generalizes relatively well, its loss on training examples is likely to be systematically lower than the loss on similar examples *not* in the training data. Given a target model $f$ trained on data assumed to be sampled i.i.d. from a distribution $P$, a typical MIA operates by training proxy models using a similar architecture and training procedure on "background" data sampled from $P$ but known to be disjoint from the training data of $f$.

In practice, an auditor or adversary is unlikely to have sample access to a distribution that is identical to $P$ and disjoint from the training data of $f$. Consider, for example, the real-world AI safety scenario of detecting whether child sexual abuse material (CSAM) was used in a model's training data (Thiel, 2023; Thiel et al., 2023; Kapoor et al., 2024; Thorn and All Tech is Human, 2024). Here, MIAs could be useful to flag potentially offending models for further scrutiny (Thorn and All Tech is Human, 2024). However, due to legal and ethical restrictions, auditors are often unable to directly access examples of CSAM themselves to train proxy models for membership inference (Thiel, 2023).

This example is part of a broader set of applications that we formalize in this work, in which an auditor or adversary wants to query membership of samples from some class $i$ in the data, but has *no samples* from class $i$ when training the attack. This is a new data access model not previously studied in the MIA literature, which we call the *"unseen class"* setting. This setting captures not only the CSAM auditing scenario but also other safety- or resource-critical auditing scenarios: for example, where the target training data may be proprietary or sensitive (such as auditing models trained on medical records) or where attack training data may be scarce (e.g. if a third party is auditing a large company's model).

To perform a membership inference attack, the state-of-the-art approach is to train a large number of *shadow models* (Carlini et al., 2022) (models that aim to mimic the behavior of the target model

but are trained on controlled subsets of data) in order to model the target model's behavior when a given sample is or is not in the training data. This attack trains several models on nonmember data to solve the same task as the target model, and then uses these as proxies to decide whether the point is more likely to be a member or nonmember in the target.

In this work, we show that shadow model-based attacks can fail catastrophically when performing membership inference in the unseen class setting. Shadow models crucially rely on training models that *perform the same task as the target model*, and with no samples from class $i$, a classifier may assign zero probability to label $i$, causing the classification task to fail on samples from the unseen class. To our knowledge, we are the first to identify a significant failure mode of shadow model-based attacks, which are generally considered to be the gold standard for MIA (Carlini et al., 2022; Zarifzadeh et al., 2023).

The failure of shadow model-based attacks shows that the unseen-class setting is challenging. However, we identify that a simple, computationally efficient baseline—*quantile regression* attacks (Bertran et al., 2023; Tang et al., 2023)—significantly outperforms shadow model attacks in this scenario. Quantile regression attacks were originally proposed as an MIA that can provide a provable guarantee on the attack's false-positive rate by predicting an $\alpha$-quantile of the nonmember score distribution. We observe that quantile regression attacks also *generalize* well to unseen classes, because they learn features that correlate with score across classes. This gives them an unexpected advantage in the unseen-class setting.

We verify this intuition both theoretically and empirically, using a set of benchmarks with image, language, and tabular data to investigate the unseen class setting in practice.

Overall, we make the following contributions:

- We identify a new MIA data access model, the unseen-class setting. This setting captures concerns in real-world auditing where data access is limited due to legal, ethical, or logistical constraints, such as CSAM detection, where the target data is not available when the attack models are being trained.
- We show that, in this setting, the most popular and state-of-the-art approaches based on training shadow models deteriorate in performance—on par with or worse than a simple baseline based on global thresholding.
- We evaluate a baseline based on *quantile regression* attacks and find surprisingly strong results in the unseen-class setting across data domains:
  - On image data, in the 1% FPR regime, quantile regression can achieve up to **11×** the TPR of shadow models on the unseen class (on CIFAR-100). Meanwhile, on ImageNet, we find that quantile regression can achieve 3.8% TPR at 1% FPR (about half the TPR achieved by full training) with access to only *10% of training classes.*
  - On tabular data, in the 1% FPR regime, quantile regression achieves up to 2× the TPR of shadow models and improves AUC by 10 points.
  - Attacking a GPT-2 classifier, quantile regression achieves 6× the TPR of shadow models in text classification (20 Newsgroups) in the 1% FPR regime.
- Finally, we provide a theoretical model illustrating the benefits and potential limitations of quantile regression in this setting. Our analysis helps to better explain the effectiveness of the approach and also points to several directions of future study.

The unseen-class problem setting captures important properties of real-world privacy auditing that have not been captured by previous MIA threat models. This setting is challenging, but our evaluation shows that quantile regression is a promising approach due to its generalization properties. We hope that these initial results inspire the community to develop further MIAs that address the real constraints of privacy auditing in highly sensitive settings.

## 2 BACKGROUND AND PRELIMINARIES

We first formalize the membership inference attack (MIA) setting and introduce relevant attack methods.

We begin with the supervised learning setup. Let $\mathcal{D} \in \Delta(\mathcal{X} \times \mathcal{Y})$ denote the data distribution over input features $\mathcal{X}$ and labels $\mathcal{Y}$. The target model $f$ is trained on a dataset $D_{\text{priv}} \sim \mathcal{D}$ consisting of $n_{\text{priv}}$ labeled examples $(x_i, y_i)$. In the classification setting, we assume $\mathcal{Y}$ is a finite label set with $|\mathcal{Y}| = c$. The model $f$ outputs a vector of logits, i.e., $f : \mathcal{X} \to \mathbb{R}^c$.

In a membership inference attack, the adversary aims to determine whether a given *target* example $(x,y)$ was part of the private training dataset $D_{\text{priv}}$. The adversary is typically assumed to have access to an auxiliary dataset $D_{\text{pub}} \sim \mathcal{D}$, which is disjoint from the private dataset, i.e., $D_{\text{pub}} \cap D_{\text{priv}} = \emptyset$. However, in this work we consider a practical setting where $D_{\text{pub}}$ is drawn from a more restricted sub-population.

**Setting: MIA with limited access to classes.** In practice, the adversary may only have access to samples from a *subset* of the classes used to train the target model. This can happen for a number of reasons. For example, the background data may be drawn from a public source such as data available on the Internet, while the target model may include samples from private or proprietary data sources. For auditors who want to run MIA to audit a model for potentially harmful content (as in the CSAM example described above (Thiel, 2023)), that content may not be legally available to the auditor at large enough scale to train the attack. Alternatively, the adversary may simply be resource-limited and unable to collect representative samples covering the space of data used to train the target model.

Let $Y_d \subseteq \mathcal{Y}$ denote the set of *unseen* classes. In this setting, the adversary has access to a public dataset $D'_{\text{pub}}$ drawn from the conditional distribution $\mathcal{D}_{\neg Y_d} := \mathcal{D} \mid y \notin Y_d$, which contains only *seen* classes. Despite this restriction, we aim to evaluate the performance of the membership inference attack (MIA) on target examples drawn from the full distribution $\mathcal{D}$. This setup allows us to study whether MIA methods trained on a restricted subset of classes can generalize to previously unseen classes from the original distribution.

We now introduce three classes of MIA methods. Each method relies on a *score function $s$* that assigns a numeric score to a target example $(x,y)$, intended to reflect the likelihood that $(x,y)$ was included in the training set for the model $f$. For example, in Bertran et al. (2023); Carlini et al. (2022), an example of such a score function is based on logit differences:

$$s(x,y,f) = f(x)_y - \max_{y' \neq y} f(x)_{y'} \tag{1}$$

**1) Marginal baseline attack with a single threshold.** The marginal baseline attack(Yeom et al., 2018) (which we refer to as "LOSS" throughout the paper in keeping with (Carlini et al., 2022)) is a simple yet widely used baseline for membership inference. Here, the attacker chooses a single threshold $\tau$ such that any example $(x,y,f)$ with $s(x,y) > \tau$ is predicted to be a member of $D_{\text{priv}}$, and otherwise it is predicted to be a non-member. In our setting, we compute this threshold based on the held-out public dataset $D'_{\text{pub}}$. Specifically, the threshold $\tau$ is chosen to control the false positive rate (FPR) over $D'_{\text{pub}}$. Note that the FPR computed over $D'_{\text{pub}}$ may not accurately reflect the FPR under the distribution $\mathcal{D}$ due to the distribution shift.

**2) Shadow model-based attacks.** Unlike the simple marginal baseline attack, shadow model attacks consider performing MIA with per-example thresholds. In particular, each shadow model (Shokri et al., 2017; Carlini et al., 2022) constructs a reference distribution over model outputs to evaluate membership of a target example. Formally, the adversary trains $k$ shadow models $g_1,...,g_k$, each solving the same classification task as the target model $f$, with architecture and training procedure identical to that of $f$, and trained using the same algorithm as $f$. Most SoTA attacks use such shadow models as a core component of the attack (Carlini et al., 2022; Choquette-Choo et al., 2021; Zarifzadeh et al., 2023).

In our setting, each shadow model is trained on random subsets drawn independently from the public dataset $D'_{pub}$. In keeping with the computationally tractable "offline" attack in Carlini et al. (2022), for each target point, we only use the set of shadow models for which the training point is a *nonmember*. [1] The collection of shadow models allows the attacker to learn the conditional distribution of the scores:

$$P_{\text{out}} : s(x,y,g) \mid (x,y) \text{ is not used in training } g$$

Given the target model $f$, the attacker decides membership based on the probability of the score $s(x,y,f)$ under $P_{\text{out}}$.

**3) Quantile regression attack.** The quantile regression attack (Bertran et al., 2023; Tang et al., 2023) offers an efficient alternative to shadow model approaches for membership inference. Rather than fitting a distribution over shadow model outputs, the attacker directly learns a function that maps

---

[1] Carlini et al. (2022) evaluate both the online and offline attacks and show that the difference in the ROC curves is minimal.

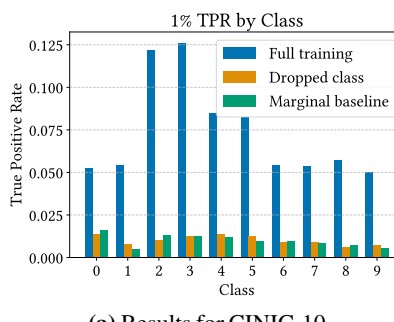 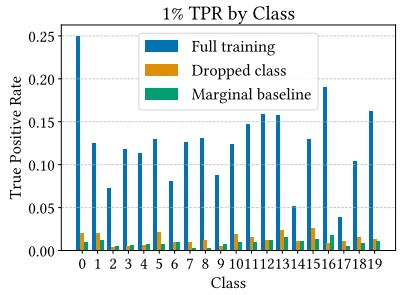

**(a)** Results for CINIC-10.  **(b)** Results for CIFAR-100 (coarse labels).

**Figure 1:** True positive rates for shadow model attacks in the 1% false positive rate regime for CINIC-10 and CIFAR-100 (we defer the 0.1% regime to Appendix B). Each bar represents the TPR on the indicated class. "Full training" refers to the TPR on class $i$ when no classes are excluded from shadow model training. In yellow, we plot the TPR when that class is excluded from shadow model training. The attack success degrades significantly under class exclusion, often performing worse than the marginal baseline (global threshold).

input examples to score thresholds—thereby enabling per-example thresholds. Given a target FPR $\alpha$, the attacker trains a model $q_\alpha : \mathcal{X} \times \mathcal{Y} \to \mathbb{R}$ to estimate the $(1-\alpha)$-quantile of the score distribution, conditioned on the input $(x,y)$. The model $q_\alpha$ is trained via minimizing the *pinball loss* over a function class $\mathcal{H}$ on the public dataset

$$q_\alpha \in \arg\min_{q' \in \mathcal{H}} \mathbb{E}_{(x,y) \sim D'_{\text{pub}}}[\text{PB}_{1-\alpha}(q'(x), s(x,y,f))] \qquad (2)$$

where $\text{PB}_{1-\alpha}$ is defined as $\text{PB}_{1-\alpha}(\hat{s}, s) = \max\{\alpha(\hat{s}-s), (1-\alpha)(s-\hat{s})\}$. This loss function is well known to elicit quantiles, in the same way that squared loss elicits means. Then the MIA predicts membership if $s(x,y,f) > q_\alpha(x)$.

Notably, this approach requires training only a single, lightweight model unlike shadow model attacks, which often demand replicating the full training pipeline and architecture of the target model.

**Evaluation metrics.**  Across our main results, we use *true positive rate at low false-positive rate* as our main metric, where low false positive rates are 1% and 0.1%. This is in keeping with best practices recommended by Carlini et al. (2022).

## 3    SHADOW MODELS FAIL ON UNSEEN CLASSES

We first show that attacks that use shadow models fail in the unseen-class setting. Shadow models underpin the most popular and rigorous membership inference attacks, from the first proposals for membership inference (Shokri et al., 2017) up to the current state of the art (Zarifzadeh et al., 2023).

**Setup.**  We assume a setting where the target model is trained on half the training data, the adversary has access to all but one class (from the remaining training data) to train the attack, and the query points are drawn from the unseen class.

For each attack, we train 16 shadow models using the "offline" variant of the LiRA algorithm with a fixed, global variance estimate (following Carlini et al. (2022)). We use the LiRA attack unmodified except that the shadow models' training data excludes examples from the unseen class.

We train shadow models on CINIC-10 and CIFAR-100 (using the superclass label set consisting of 20 classes). In Figure 1 we report the performance of shadow models on queries from each class before and after dropping a single class at a time. We find that for both datasets, shadow models perform on par with a much weaker marginal baseline that does not learn per-example thresholds but rather fits a single threshold across all classes.

Intuitively, a model that does not see a class at training time will assign zero probability to that class label. As a result, the shadow models' confidences for the true label on the missing class will be zero, resulting in many false positives at test time—any higher confidence reported by the target model would be considered significant. This applies to both the LiRA attack as well as other more recent attacks (such as RMIA (Zarifzadeh et al., 2023)) that judge membership using the score of the query

on the target model *relative to* the score on reference models that have never seen the query class. In the next section, we demonstrate a similar failure for RMIA.

For completeness, in Appendix B we also include shadow model results using the difference between the top two logits (rather than the correct label) as the score function. We find that this score function reduces the baseline performance of the attack. Thus, for the remainder of our comparisons, we use the true label confidence as the score function for shadow models.

# 4 QUANTILE REGRESSION ATTACKS FOR UNSEEN CLASSES

Shadow models are fundamentally constrained because they must solve the same learning problem as the target model in order to estimate the score distribution of the target model on a given example. Quantile regression attacks (Bertran et al., 2023) were originally proposed as an MIA that can provide a provable guarantee on the attack's false-positive rate by predicting an $\alpha$-quantile of the score distribution.

In this section, we make a novel observation – that quantile regression attacks also *generalize* well to unseen classes, because they learn features that distinguish members from nonmembers. Unlike shadow model-based attacks, which judge membership by evaluating the score of the target model relative to the score of a reference model, quantile regression attacks directly predict thresholds from target model scores on the background distribution. This gives them an unexpected advantage in the unseen-class setting.

Our results give a new perspective on the strengths and weaknesses of shadow models and indicate that attacks that learn membership predictors, including but not limited to quantile regression attacks, may be much stronger candidates for *real-world* membership inference settings where the data to query is sensitive and not available for training.

## 4.1 UNSEEN CLASSES FOR IMAGE DATASETS.

Our first set of results is on image classification models. This setting models scenarios such as detecting child sexual abuse material (CSAM) in image models.

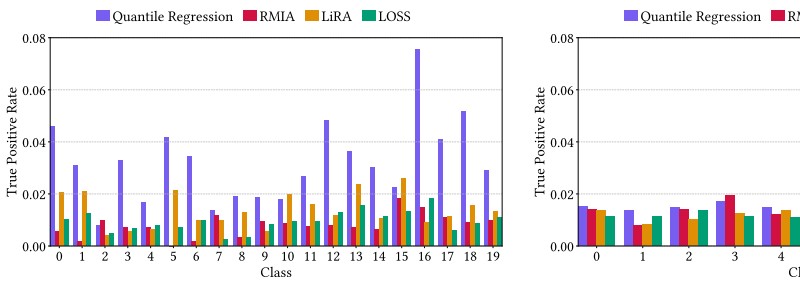

**(a)** TPR for the dropped out superclass, CIFAR-100          **(b)** TPR for the dropped out class, CINIC-10

**Figure 2:** True positive rates in the low false positive regime for CINIC-10 and CIFAR-100 (superclass set) on each unseen class. Each bar represents the true positive rate on class $i$ when class $i$ is dropped from the attack training set. We only report results at 1% FPR; the results at 0.1% FPR are not meaningful due to the small sample size of the validation set on a single class (1000 samples). While quantile regression attacks have only a small advantage over shadow models on CINIC-10 (see Figure 5), they achieve up to $11\times$ higher TPR than shadow models on CIFAR-100.

**Setup.**    We first study an identical setting where the target Resnet50 base model is trained on half the training data, the adversary has access to all but one class from the remaining training data to train the attack, and the query points are drawn from the unseen class.

As baselines, we use the LiRA attack described above, as well as the state of the art RMIA attack (Zarifzadeh et al., 2023). [2]

---

[2]Notably, in addition to training reference models, the RMIA attack also uses samples from the background distribution when *evaluating* the score function at query time. It is generally unrealistic to assume that the adversary has access to enough unseen-class, nonmember samples to perform this evaluation. However, in order to give RMIA the most advantage, we *include* unseen class samples at evaluation time (but not when training reference models, as is the case for shadow models and quantile regression).

For each attack, we train a single quantile regression model on the remaining training data, excluding the unseen class and keeping the validation set as heldout public data for evaluating FPR. Pinball loss is notoriously difficult to minimize, so for training stability, we follow Bertran et al. (2023) and train the network to fit a Gaussian (mean and variance) conditioned on each sample instead of directly using pinball loss to predict quantiles.

Our final models for the CINIC-10 and CIFAR-100 attacks are ConvNext-Tiny-224 models trained for 30 epochs, with the Adam optimizer, batch size 16, and learning rate of 1e-4. We find that early stopping does not improve the attacks. To make our attack agnostic to the true label, we modify the attack from Bertran et al. (2023) and use the difference between the top two logits as our score function:

$$s(x,y,g) = \max f(x) - \max_{y' \neq \max f(x)} f(x)_{y'}.$$

Using this score function improves quantile regression performance in the class dropout setting as the learned attack no longer requires knowledge of the true label.

**Class dropout: CINIC-10 and CIFAR-100.**  We find that for both CINIC-10 and CIFAR-100 (using the superclass label set), quantile regression strictly outperforms the marginal baseline and shadow models under class dropout at 1% FPR. In this setting, we train the attack model on all classes except the unseen class. The TPR and FPR are evaluated on the held-out class.

Results are similar on CIFAR-100 but even more pronounced. On average across all 20 superclasses, LiRA achieves only 1.4% TPR at 1% FPR on unseen data while quantile regression achieves 3.8% TPR at 1% FPR (a 2.7× improvement). In the unseen class setting, quantile regression takes less time to train and outperforms LiRA. We find that RMIA often performs *worse* than the LiRA attack. This is because RMIA's score function exaggerates the target model's confidence on the unseen class, resulting in more false positives in this setting and very low TPRs in the low-FPR regime.

These trends also hold on ResNet-18 and Vision Transformer base models; detailed results are found in Appendix E.

**Data scarcity: ImageNet.**  Our results on CINIC-10 and CIFAR-100 are limited to a single unseen class at a time, which models the impact of a minority subpopulation (such as sensitive or harmful content) missing from the attack data. Another realistic scenario where the auditor or adversary might not have access to subclasses is the *data scarcity* setting, where the model might be trained using a much larger and potentially proprietary dataset while a resource-limited auditor only has a fraction of similar data available.

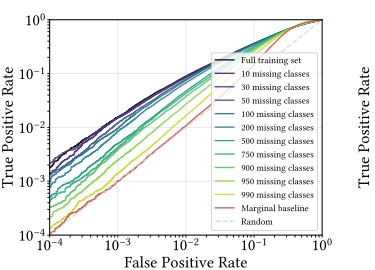 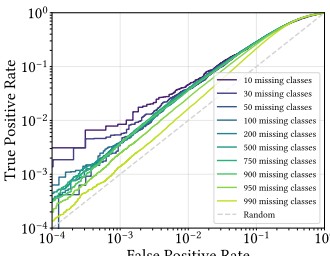 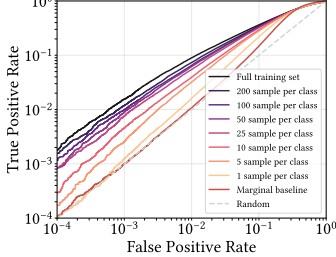

**(a)** ROC curve for class drop experiment on ImageNet.

**(b)** Unseen class ROC curve for class drop experiment on ImageNet.

**(c)** ROC curve for sample drop experiment on ImageNet.

**Figure 3:** ROC curves for class and sample drop experiments on ImageNet. Enlarged versions of the plots are provided in Appendix H.

To simulate this, we attack a model trained on the much larger ImageNet dataset. We use the same quantile regression architecture and hyperparameters as described for CINIC-10 and CIFAR-100. In Figure 3a, we show the ROC curves for ImageNet (on the full data distribution) with a sweep from 10 to 990 classes missing from the attack training set. Perhaps surprisingly, the quantile regression attack outperforms the marginal baseline with as many as 990 out of 1000 classes left unseen.

One might ask whether this effect is simply due to averaging over both missing and in-distribution classes. In Figure 3b, we plot attack performance for the missing classes alone. The performance on unseen classes remains fixed around 3.8% TPR at 1% FPR (0.4% TPR at 0.1% FPR) when anywhere

from 100 to 900 classes are removed, and remains significantly above the marginal baseline even with 990 classes removed.

Finally, another realistic setting of data scarcity is one where examples from all classes are available, but where the auditor only has very few samples from each class. We also evaluate ImageNet in this scarce-sample setting where only $k$ samples are retained from each class. Quantile regression outperforms the marginal baseline (Figure 3c) even when training data is severely limited, retaining 3.9% TPR at 1% FPR given as few as 10 samples per class (compared to 9.0% TPR at 1% when trained on all the data; on average, 401 samples per class).

We present similar results for data scarcity on CIFAR-10 and CIFAR-100 in Appendix D. Quantile regression achieves, for example, 3% TPR at 1% FPR even when half of the classes are dropped from CIFAR-100.

### 4.2  UNSEEN CLASSES FOR TABULAR AND TEXT DATASETS.

The unseen class scenario also arises in tabular and text settings. For example, consider auditing medical records in which subsets of records may be inaccessible to the auditor due to privacy, legal, or ethical restrictions (Tramèr et al., 2022), or population survey data in which some records might belong to sensitive minority groups and cannot be made public (Steed et al., 2024).

Next, we show that shadow models and global thresholding fail on unseen classes in tabular and text classification settings as well. For these experiments, we compare quantile regression to RMIA, the state-of-the-art shadow model approach from Zarifzadeh et al. (2023).

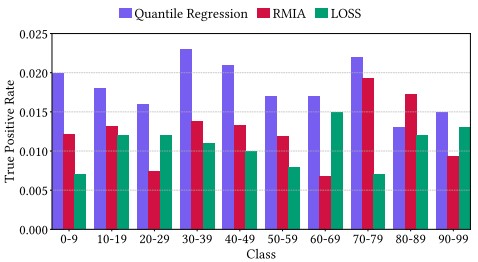 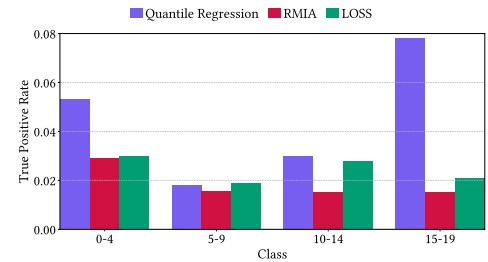

**(a)** TPR for the dropped out classes, Texas  **(b)** TPR for the dropped out class, 20 Newsgroups

**Figure 4:** True positive rates in the low false positive regime for Texas (tabular) and 20 Newsgroups (text) on sets of unseen classes. Each bar represents the true positive rate on classes $C$ when $C$ are dropped from the attack training set. We only report results at 1% FPR; the results at 0.1% FPR are not meaningful due to the small sample size of the validation set on a single class. Quantile regression attacks achieve up to $2\times$ higher TPR than shadow models on Texas and up to $6\times$ higher TPR on 20 Newsgroups.

**Setup.**  For text classification, we fine-tune GPT-2 (Radford et al., 2019) for 20-NewsGroups classification. Inputs are tokenized with truncation and padding to a fixed length of 256 tokens. We train with AdamW, batch size 32, learning rate 1e-3, weight decay 5e-4, 500 warmup steps, for 30 epochs. For Texas tabular classification, we follow Zarifzadeh et al. (2023) and train a 2-layer MLP base model. For text and tabular, our attack models are also 2-layer MLPs (hidden sizes 256, 128). For text, the input is an embedding bag over the 256 input tokens. The hyperparameters and score function match the image auditing attacks described above.

**Class dropout: Texas and 20-NewsGroups.**  We find that on tabular and text data, quantile regression again strictly outperforms the marginal baseline and RMIA shadow models under class dropout at 1% FPR. The same is true for AUC, and full results can be found in Appendix C. We drop 10% of classes for both datasets (10/100 for Texas, 2/20 for 20-NewsGroups). As described above, the TPR and FPR are evaluated on the held-out class.

### 4.3  DEFENSES

In general, models that one would wish to audit are *not* deployed with typical MIA defenses such as differential privacy (Thorn and All Tech is Human, 2024). Nevertheless, for completeness, we evaluate our method in the unseen-class setting on models trained with one of two defenses: weight decay (equivalent to $\ell_2$ regularization for SGD) and differential privacy. We note that differential privacy has been noted by several previous works to be a strong (provable) MIA defense that prevents most MIA

methods from succeeding, but is highly impractical (Leino and Fredrikson, 2020; Choquette-Choo et al., 2021; Carlini et al., 2022; Li and Zhang, 2021) due to the difficulty of achieving reasonable test accuracy on models trained from scratch.

We provide these results in Appendix F. Predictably, we find that weight decay is less effective than DP at preventing MIA from succeeding and more regularization leads to worse MIA success. In reasonable defense regimes where the TPR at low FPR exceeds random guessing, quantile regression still outperforms the baseline in the unseen class setting.

## 5 THEORETICAL MODEL

Our empirical results show that the quantile regression attack can achieve nontrivial accuracy even under extreme data scarcity when training the attack model. However, it is not clear from our empirical results *when* we might expect quantile regression to succeed.

As a step toward understanding our results, we prove a "transferability" theorem for quantile regressors. Intuitively, this theorem states that if the distribution of sample embeddings under the quantile regression model with and without the unseen classes is "similar," then the FPR guarantee of the quantile regressor trained on only the seen classes should also hold on the full query set.

**Definition 5.1** (Pinball loss). For a quantile level $\alpha \in (0,1)$, the *pinball loss* (also known as the check loss) for a prediction $\hat{s}$ and true outcome $s$ is defined as:

$$\ell_\alpha(\hat{s},s) = (\alpha - \mathbf{1}\{s < \hat{s}\})(s - \hat{s}) = \begin{cases} \alpha(s - \hat{s}) & \text{if } s \geq \hat{s}, \\ (1-\alpha)(\hat{s}-s) & \text{if } s < \hat{s}. \end{cases}$$

**Quantile Regression Predictor.** We consider a class of linear quantile regression predictors:

$$q_\alpha(x) = \langle \phi(x), w \rangle,$$

where $\phi : \mathcal{X} \to \mathbb{R}^d$ is a fixed feature mapping, and $w \in \mathcal{W}$ is a weight vector. For any distribution $P$ over $\mathcal{X} \times \mathcal{S}$, we will write $P_\phi$ to denote its induced distribution over $(\phi(x), s)$.

We will focus on the case where $\mathcal{W} = \mathbb{R}^d$, but should generalize to more constrained set of weights later. We adapt the multi-accuracy definition (Roth, 2022; Hebert-Johnson et al., 2018) to our specific setting with a feature mapping.

**Definition 5.2** (Multi-Accuracy for Quantile Prediction). A predictor $q_\alpha : \mathcal{X} \to \mathbb{R}$ is said to be $(\mathcal{W}, \phi, \varepsilon)$-*multi-accurate* for quantile level $\alpha$ with respect to distribution $P$ if, for every $w \in \mathcal{W}$,

$$\left| \mathbb{E}_{(x,s) \sim P}[\langle w, \phi(x) \rangle \cdot (\mathbf{1}\{s < q_\alpha(x)\} - \alpha)] \right| \leq \varepsilon.$$

We now show that multi-accuracy, when instantiated for quantile prediction, provides a sufficient condition for calibration to transfer across distributions. We consider the setting where a quantile predictor is trained on distribution $P$ and deployed on a shifted distribution $Q$. The theorem below shows that if the feature representation captures the density ratio between $P$ and $Q$ via a linear function, then the learned predictor remains calibrated at the target quantile level under $Q$. This result can be viewed as a specialized instance of the *universal adaptability* framework of Kim et al. (2022), tailored to multi-accurate quantile predictors derived from empirical risk minimization.

**Theorem 5.3** (Transferability of Quantile Predictors). *Let $P$ and $Q$ be distributions over $(x,s)$, and let $\phi : \mathcal{X} \to \mathbb{R}^d$ be a fixed feature map. Suppose we learn a linear quantile predictor $q_\alpha(x) = \langle \phi(x), w^* \rangle$ by minimizing the expected pinball loss under $P$:*

$$w^* \in \arg\min_{w \in \mathcal{W}} \mathbb{E}_{(x,s) \sim P}[\ell_\alpha(\langle \phi(x), w \rangle, s)].$$

*Assume that the density ratio between $Q$ and $P$ satisfies:*

$$\frac{dQ_\phi(\phi(x),s)}{dP_\phi(\phi(x),s)} = \langle \phi(x), v \rangle \quad \text{for some } v \in \mathcal{W}, \text{ and for all } (\phi(x),s) \in \text{supp}(Q_\phi).$$

*Then the learned predictor $q_\alpha$ is calibrated under distribution $Q$ at quantile level $\alpha$:*

$$\mathbb{E}_{(x,s) \sim Q}[\mathbf{1}\{s < q_\alpha(x)\} - \alpha] = 0.$$

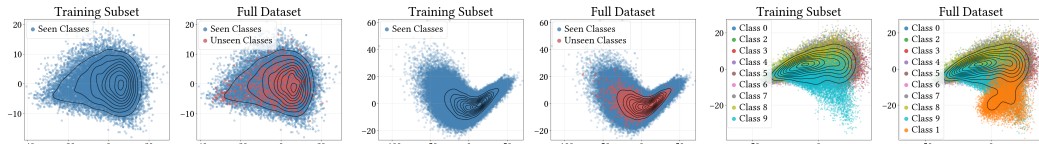

**(a)** Gaussian mixtures fit to CIFAR-100 data with one unseen class.

**(b)** Gaussian mixtures fit to Imagenet data with ten unseen classes.

**(c)** Gaussian mixtures fit to CINIC-10 data with one unseen class.

**Figure 5:** Visualization of Gaussian mixture models fit to the (dimension-reduced) embeddings learned by the quantile regression models trained on on subsets of CINIC-10, CIFAR-100, and Imagenet. Dropping a class largely does not change the distribution over embeddings for CIFAR-100 and Imagenet, where we observe that quantile regression is the most effective.

We defer the proof to Appendix G.

Intuitively, the transferability theorem states that when there exists a linear transformation between the feature representation of the (unseen-class) training distribution and the feature representation of the full distribution, the false-positive-rate guarantee of the quantile regressor over the training distribution will also hold for the test (full) distribution. The assumption of optimality in the last layer is mild, since the pinball loss is convex in $w$ given fixed features.

### 5.1 Empirical Estimates

To understand the implications of this theorem in our setting, we can treat the quantile regressor as a feature extractor using the final layer before the prediction step to generate embeddings $\phi(x)$ for each sample in the dataset. We compare the distribution over last-layer embeddings in the attack training set (the seen subset of classes) with the distribution over last-layer embeddings in the evaluation set.

Thus, in order to show that the theorem statement holds, we need to measure the density ratio between the attack training distribution and the test distribution. In practice, we approximate the density ratio using Gaussian mixture models trained over a low-dimensional projection of the embeddings (the top two PCA components) rather than the original embeddings due to the relatively low sample size. These Gaussians are visualized in Figure 5.

Using these approximations, we explicitly measure the density ratio over the test set and fit a linear layer on top of the representation learned by the quantile regressor. We find that the linear fit improves as the dataset size and diversity increases: the MSE of the linear model on CINIC-10 is 8.11e-3, on CIFAR-100 is 2.10e-3, and on Imagenet 1.85e-3. This experiment provides some validation of the empirical results we see in practice and may expect based on the visual representation of the image embeddings: the results of quantile regression are weakest on CINIC-10 (where the linear fit is comparatively worse) and best on ImageNet (where the linear fit is best).

### 6 Related Work

**Membership inference attacks.** Shadow models were initially proposed by Shokri et al. (2017) and refined by Carlini et al. (2022). Membership inference attacks based on quantile regression were introduced by Bertran et al. (2023) for classification tasks. Follow-up work (Tang et al., 2023; Carlini et al., 2023) extended this approach to the diffusion model setting using a score metric inspired by Duan et al. (2023). Others (Zhang et al., 2024; Hayes et al., 2025) have made progress towards applying these methods to large language models.

A number of earlier works propose using a single global score threshold across examples rather than fitting per-example thresholds (Shokri et al., 2017; Yeom et al., 2018; Song et al., 2019; Duan et al., 2023). Later work (Watson et al., 2021; Carlini et al., 2022) has shown that these global score thresholds perform poorly compared to thresholds calibrated to example difficulty, particularly in the low-false positive rate regime.

**MIA under distribution shift.** A number of prior works have studied the effectiveness of shadow model attacks under varying distribution shift, although none of these study the unseen class setting that we study in our work. Yichuan et al. (2024) study a setting in which the attacker has access to data encompassing a *superset* of the target labels, as well as attribute shift corresponding to subpopulation reweighting. Liu et al. (2022) study distribution shift between CIFAR-10 and CINIC-10, which have the same label set. Earlier works (Carlini et al., 2022; Shokri et al., 2017) have empirically studied how shadow models perform when the target architecture or training procedure are not known exactly.

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

## A  EXPERIMENTS AND IMPLEMENTATION DETAILS

Shadow models were trained on 2 NVIDIA A100 GPUs using the code released by Carlini et al. (2022). The code was used unmodified except to drop the relevant classes from the attack model training data.

The quantile regression models were trained on 4 A800 nodes, taking about 3000MB per node to train ConvNext-Tiny-224 models. Each model took as input a 224x224x3 image and returned 2 outputs, the predicted Gaussian mean and variance. Each model was trained for 30 epochs. For CINIC-10, training with no class dropout took approximately 40 minutes per model. For CIFAR-100, training took approximately 12 minutes per model. For ImageNet, training took approximately 5 hours per model. Under class dropout and sample dropout, the training dataset was smaller and training time reduced.

We also experimented with running ConvNext-Large-224, which took 11000MB per node to train, and significantly longer, i.e. 4 hours per model for CINIC-10.

## B  ADDITIONAL SHADOW MODEL RESULTS

We provide additional results in the FPR 0.1% regime to supplement Figure 1.

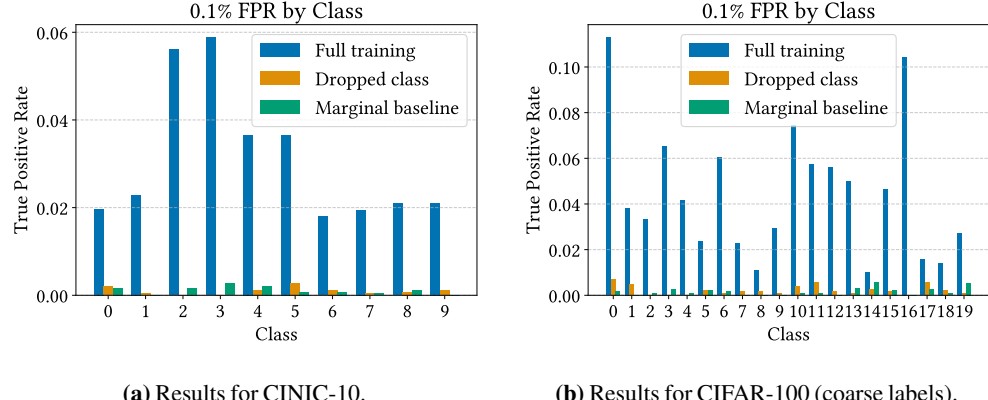

**(a)** Results for CINIC-10.                    **(b)** Results for CIFAR-100 (coarse labels).

**Figure 6:** True positive rates for shadow model attacks in the 0.1% false positive rate regime for CINIC-10 and CIFAR-100. Each bar represents the TPR on the indicated class. In yellow, we plot the TPR when that class is excluded from shadow model training. The attack success degrades significantly under class exclusion, often performing worse than the marginal baseline (global threshold).

As described in Section 3, one possible explanation for shadow models' underperformance is due to the true logit score function which fails in an unseen class setting. However, we also test the top-two logit difference as a score function, and find that shadow models perform even worse in this setting.

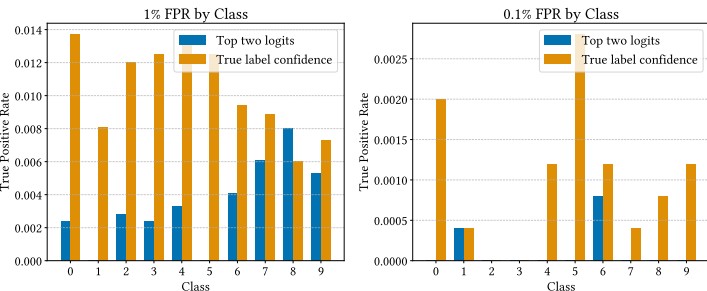

**Figure 7:** Comparison between shadow model attack success (with class $i$ dropped) with true label confidence as the score metric and top-two logit difference as the score metric on CINIC-10. Although the top-two logit difference does not use the true (dropped) label that was not seen by the shadow models, the attack performs even worse than with the true label confidence.

# C   AUC RESULTS

We provide AUC results to supplement the TPR at low FPR results provided in the paper. However, we note that AUC is *not* the best-practice metric for measuring MIA success (see, e.g., Carlini et al. (2022)) because AUC integrates over all false-positive rates. Quoting Carlini et al. (2022): "[...] the AUC is not an appropriate measure of an attack's efficacy, since the AUC averages over all false-positive rates, including high error rates that are irrelevant for a practical attack. The TPR of an attack when the FPR is above 50% is not meaningfully useful, yet this regime accounts for more than half of its AUC score." Nevertheless, we provide AUC results for completeness here.

We find that results are inconclusive for AUC as compared to the low-FPR regime, where quantile regression definitively wins across datasets and domains. On image datasets, RMIA outperforms qunatile regression on CIFAR-10, but results are inconclusive on CIFAR-100. On the Texas tabular dataset and 20 Newsgroups text dataset, quantile regression outperforms all other methods.

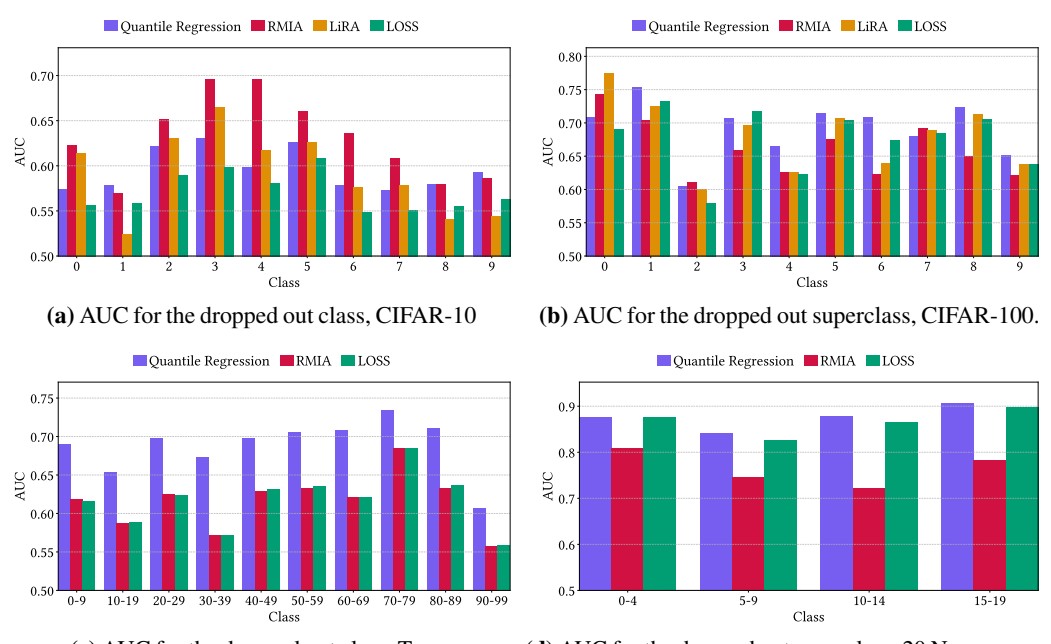

**(a)** AUC for the dropped out class, CIFAR-10

**(b)** AUC for the dropped out superclass, CIFAR-100.

**(c)** AUC for the dropped out class, Texas

**(d)** AUC for the dropped out superclass, 20 Newsgroups.

**Figure 8:** AUCs for CINIC-10, CIFAR-100, Texas (tabular) and 20 Newsgroups (text) on sets of unseen classes. Each bar represents the AUC on classes $C$ when $C$ are dropped from the attack training set.

# D CINIC-10 AND CIFAR-100 MULTICLASS DROPOUT RESULTS

We additionally provide AUC and TPR at FPR=1% for settings in CINIC-10 and CIFAR-100 where multiple classes are dropped. In this setting, quantile regression continues to outperform the LOSS baseline, achieving TPR of 3% at low FPR even when half of the superclasses are dropped from CIFAR-100. AUC scores are comparable or better for quantile regression and LOSS.

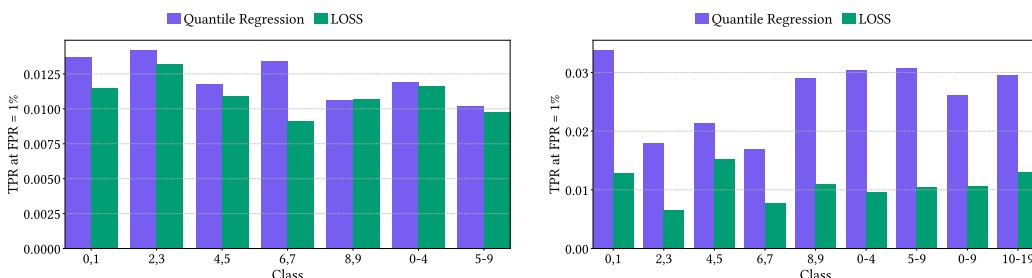

**(a)** TPR for the dropped out classes, CIFAR-10.   **(b)** TPR for the dropped out superclasses, CIFAR-100.

**Figure 9:** TPRs for CINIC-10 and CIFAR-100 on sets of unseen classes. Each bar represents the TPR on classes $C$ when $C$ are dropped from the attack training set. We only report results at 1% FPR; the results at 0.1% FPR are not meaningful due to the small sample size of the validation set on a single class.

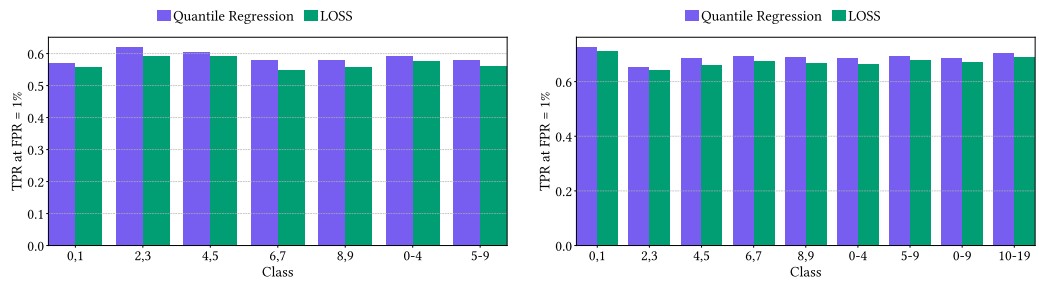

**(a)** AUC for the dropped out classes, CINIC-10.   **(b)** AUC for the dropped out superclasses, CIFAR-100.

**Figure 10:** AUCs for CINIC-10 and CIFAR-100 on sets of unseen classes. Each bar represents the AUC on classes $C$ when $C$ are dropped from the attack training set.

# E  CINIC-10 AND CIFAR-100 ALTERNATE BACKBONE RESULTS

The CINIC-10 and CIFAR-100 in the main paper audit a ResNet-50 architecture classifier, so here we show quantile regression results for ResNet-18 and Vision Transformer (`vit-base-patch16-224`). Again, quantile regression has a striking advantage at low FPR while AUC scores are comparable or better across the board.

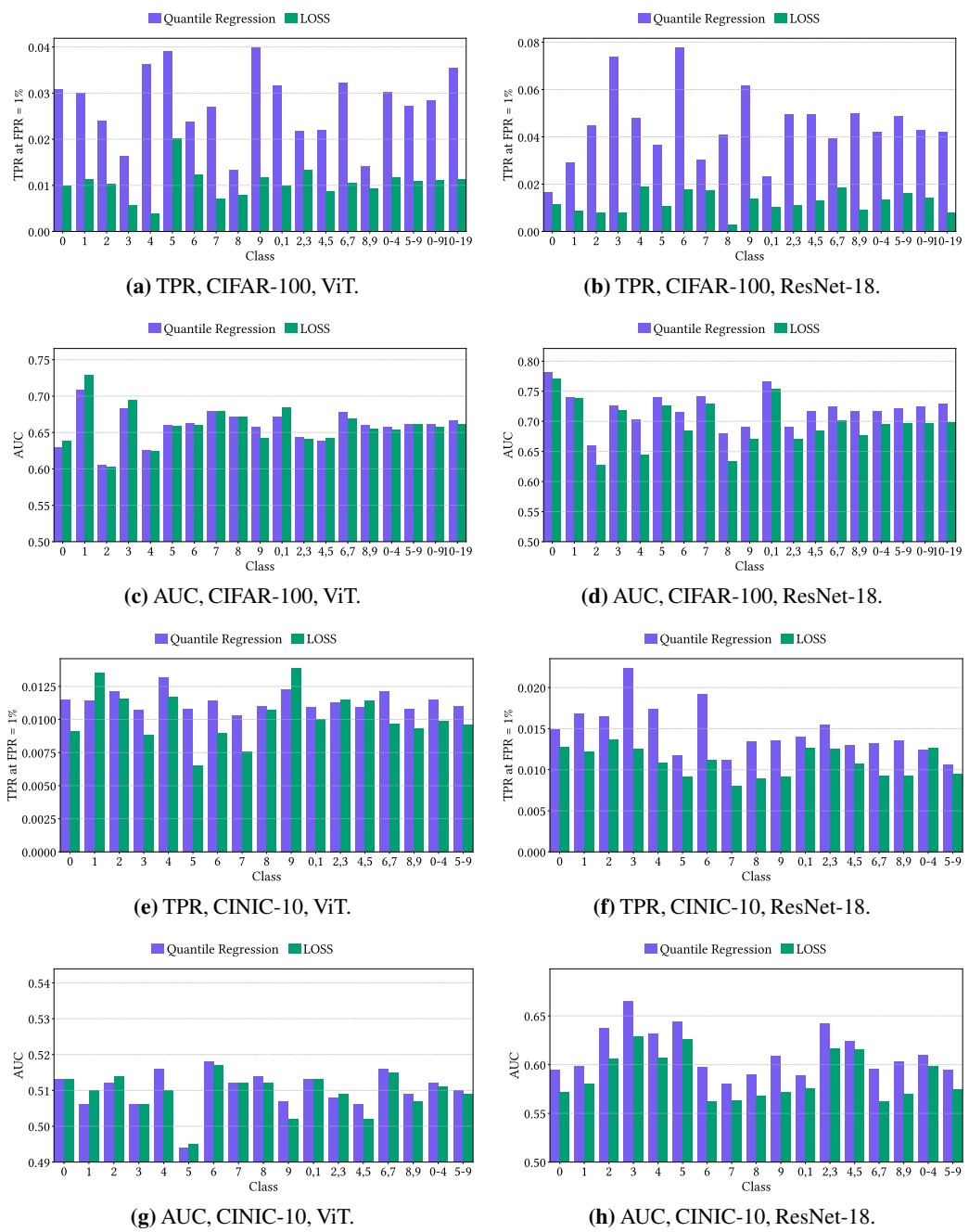

**Figure 11:** AUC and TPR for CINIC-10 and CIFAR-100 on sets of unseen classes. Each bar represents the metric on classes $C$ when $C$ are dropped from the attack training set. We only report results at 1% FPR; the results at 0.1% FPR are not meaningful due to the small sample size of the validation set on a single class.

# F RESULTS ON MODELS TRAINED WITH DEFENSES

**L2 Regularization.** For completeness, we study the unseen-class membership inference setting under common defenses. First, we measure the effect of L2 regularization, a known MIA defense, e.g., (Li and Zhang, 2021), (Choquette-Choo et al., 2021), (Leino and Fredrikson, 2020), on our method. Since our models are trained with vanilla SGD, L2 regularization is equivalent to weight decay Loshchilov and Hutter (2017), so we proceed by modifying weight decay.

**Table 1:** Comparison of MIA on CIFAR-100 under Weight Decay

|  | TPR @ FPR = 1% | | | AUC | | |
| --- | --- | --- | --- | --- | --- | --- |
| **Model** | **1cls** | **2cls** | **5cls** | **1cls** | **2cls** | **5cls** |
| QMIA (wd $= 5 \times 10^{-4}$) | **4.59** | **3.38** | **3.04** | **0.709** | **0.726** | **0.686** |
| LOSS (wd $= 5 \times 10^{-4}$) | 2.00 | 1.28 | 0.96 | 0.690 | 0.711 | 0.663 |
| QMIA (wd $= 5 \times 10^{-3}$) | **2.50** | **1.73** | **1.95** | **0.576** | **0.591** | **0.612** |
| LOSS (wd $= 5 \times 10^{-3}$) | 1.59 | 1.53 | 1.38 | 0.573 | 0.583 | 0.588 |
| QMIA (wd $= 5 \times 10^{-2}$) | **1.25** | **1.16** | 0.81 | 0.520 | 0.505 | **0.512** |
| LOSS (wd $= 5 \times 10^{-2}$) | 0.92 | 0.95 | **0.86** | **0.527** | **0.518** | **0.512** |

As L2 regularization (i.e. weight decay) increases, both QMIA and LOSS become less effective. However, the resulting model also has significantly lower test accuracy ($76\% \rightarrow 65\% \rightarrow 55\%$), so in general, sufficient L2 regularization is impractical.

**Differential Privacy (DP).** We additionally analyze unseen-class membership inference under epsilon-delta DP guarantees; however, as shown in Leino and Fredrikson (2020); Carlini et al. (2022); Li and Zhang (2021); Choquette-Choo et al. (2021), DP is an unrealistic defense when training models from scratch, as even modest guarantees require significantly degrading model performance.

In the below experiments, we train DP models from scratch on CIFAR-10 and CIFAR-100 with one class dropped out.

Following Leino and Fredrikson (2020), we evaluate DP at $\varepsilon = 1.0, 4.0, 16.0$. Our training and test accuracies on CIFAR-10 and CIFAR-100 are comparable to the results in Leino and Fredrikson (2020). We find that DP in fact prevents quantile regression (as well as the LOSS baseline) from achieving meaningful TPR or AUCs in these settings.

**Table 2:** Performance Metrics Under Differential Privacy Settings for CIFAR10 and CIFAR100

| | | | | TPR @ FPR = 1% | | AUC | |
| --- | --- | --- | --- | --- | --- | --- | --- |
| **Dataset** | $\epsilon$ | **Train Acc.** | **Test Acc.** | **QMIA** | **LOSS** | **QMIA** | **LOSS** |
| | 1 | **0.1005** | 0.0998 | 0.0056 | **0.0110** | 0.51 | 0.50 |
| CIFAR10 | 4 | **0.1029** | 0.1007 | **0.0127** | 0.0123 | 0.52 | 0.52 |
| | 16 | 0.0975 | **0.1008** | 0.0076 | **0.0100** | 0.49 | 0.49 |
| | 1 | 0.0522 | **0.0537** | 0.0025 | **0.0042** | 0.49 | 0.49 |
| CIFAR100 | 4 | **0.0509** | 0.0498 | **0.0184** | 0.0067 | 0.51 | 0.51 |
| | 16 | 0.0494 | **0.0500** | 0.0025 | **0.0109** | 0.51 | 0.50 |

# G  PROOF OF THEOREM 5.3

*Proof.* As a first step, we prove by contradiction that the learned predictor $q_\alpha^*$ is $(\mathcal{W},\phi,0)$-multi-accurate under $P$. Suppose not, then, by the definition of multi-accuracy, there exists some $w' \in \mathcal{W}$ such that

$$\mathbb{E}_{(x,s)\sim Q}[\langle w',\phi(x)\rangle \cdot (\mathbf{1}\{s < q_\alpha^*(x)\} - \alpha)] \neq 0.$$

Without loss of generality, suppose this expectation is strictly positive.

Since the pinball loss is convex and differentiable almost everywhere, its subgradient with respect to the weights at $w^*$ is:

$$\nabla_w \mathbb{E}_{(x,s)\sim P}[\ell_\alpha(\langle w,\phi(x)\rangle,s)]\big|_{w=w^*} = -\mathbb{E}_{(x,s)\sim P}[(\alpha - \mathbf{1}\{s < q_\alpha^*(x)\})\phi(x)].$$

Taking the inner product of this gradient with $w'$, we obtain:

$$\langle w', \nabla_w \mathbb{E}_P[\ell_\alpha(\langle w,\phi(x)\rangle,s)]\big|_{w=w^*}\rangle = -\mathbb{E}_P[\langle w',\phi(x)\rangle \cdot (\alpha - \mathbf{1}\{s < q_\alpha^*(x)\})] < 0,$$

by assumption. Therefore, moving in the direction $-w'$ decreases the expected pinball loss objective, contradicting the optimality of $w^*$.

Thus, we must have:

$$\left|\mathbb{E}_{(x,s)\sim P}[\langle w,\phi(x)\rangle \cdot (\mathbf{1}\{s < q_\alpha^*(x)\} - \alpha)]\right| = 0 \quad \text{for all } w \in \mathcal{W},$$

i.e., $q_\alpha^*$ is $(\mathcal{W},0)$-multi-accurate under $P$.

Finally, given that $\frac{dQ}{dP}(x)$ satisfies:

$$\frac{dQ_\phi}{dP_\phi}(\phi(x),s) = \langle \phi(x),v\rangle \quad \text{for some } v \in \mathcal{W}, \text{ with } \langle \phi(x),v\rangle > 0 \text{ for all } x \in \mathrm{supp}(Q),$$

we can perform a change of measure from $P$ to $Q$:

$$\begin{aligned}
\mathbb{E}_{(x,s)\sim Q}[(\mathbf{1}\{s < q_\alpha^*(x)\} - \alpha)] &= \mathbb{E}_{(\phi(x),s)\sim Q_\phi}[(\mathbf{1}\{s < \langle \phi(x),w^*\rangle\} - \alpha)] \\
&= \mathbb{E}_{(\phi(x),s)\sim P_\phi}\left[\frac{dQ_\phi(\phi(x),s)}{dP_\phi(\phi(x),s)}(\mathbf{1}\{s < \langle \phi(x),w^*\rangle\} - \alpha)\right] \\
&= \mathbb{E}_{(\phi(x),s)\sim P_\phi}[\langle \phi(x),v\rangle (\mathbf{1}\{s < \langle \phi(x),w^*\rangle\} - \alpha)] \\
&= \mathbb{E}_{(x,s)\sim P}[\langle \phi(x),v\rangle (\mathbf{1}\{s < q_\alpha^*(x)\} - \alpha)] = 0
\end{aligned}$$

This completes the proof. $\qquad\square$

# H ENLARGED PLOTS

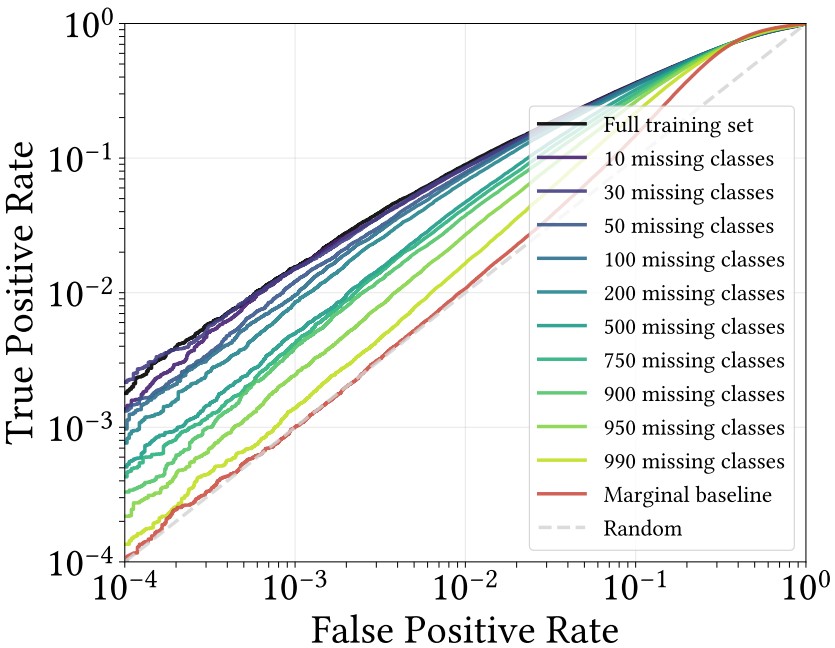

**Figure 12:** ROC curve for class drop experiment on ImageNet (Figure 3a enlarged).

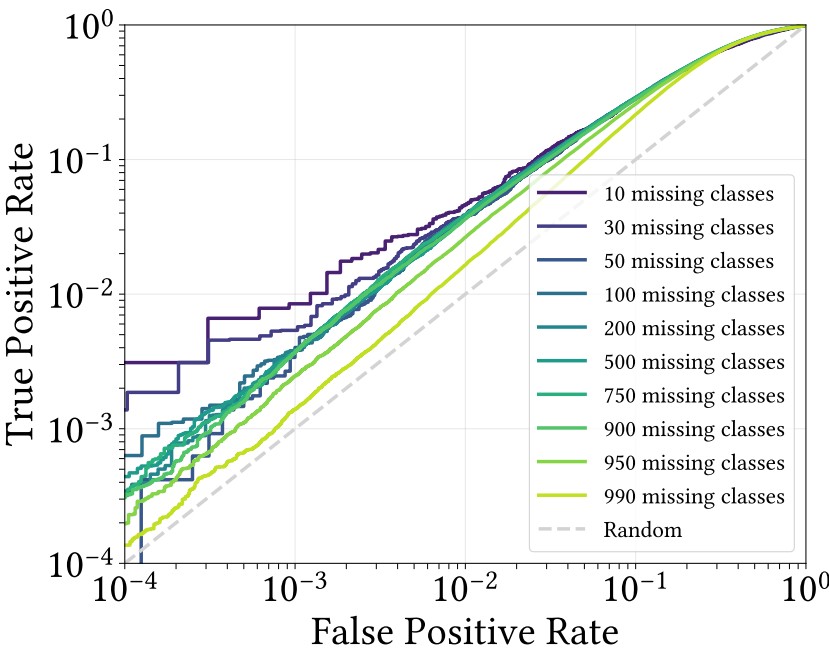

**Figure 13:** Unseen class ROC curve for class drop experiment on ImageNet (Figure 3b enlarged).

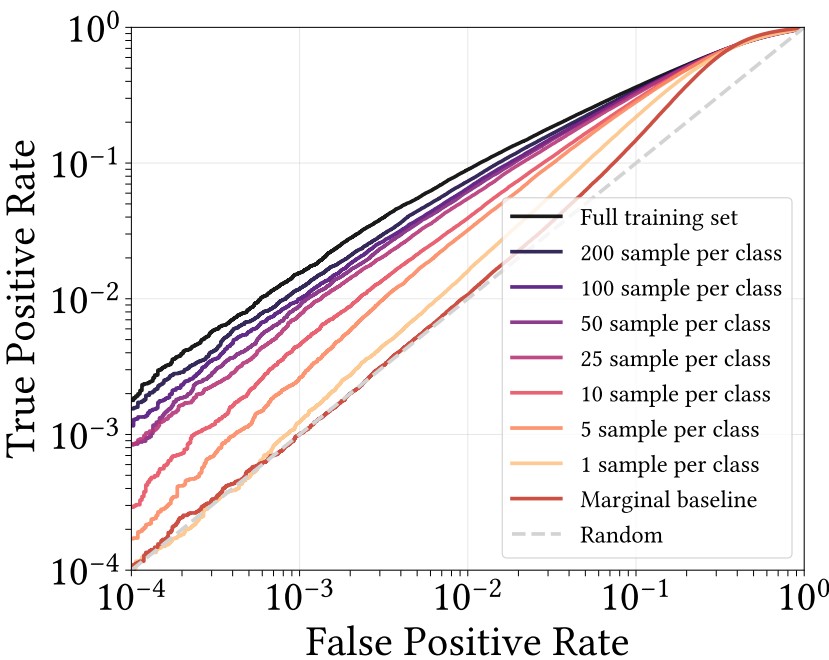

**Figure 14:** ROC curve for sample drop experiment on ImageNet. (Figure 3c enlarged).

# I   LLM USAGE

LLMs were used in writing boilerplate experiment code, debugging, and writing boilerplate plotting code. No LLMs were used for paper writing, research ideation, or related work discovery.

