# OpenReview forum: "Membership Inference Attacks for Unseen Classes"
_ICLR.cc/2026/Conference — Submitted to ICLR 2026_

### Official Review · Reviewer_C19G · 2025-10-27

**Soundness:** 2
**Presentation:** 2
**Contribution:** 2
**Rating:** 2
**Confidence:** 4

**Summary:**

This paper investigates membership inference attacks (MIAs) in a novel and challenging setting where the adversary lacks training data from certain classes—termed the "unseen class" setting. The work identifies a critical failure mode of state-of-the-art shadow model-based attacks when applied to such scenarios, particularly in high-stakes applications like detecting child sexual abuse material (CSAM) where access to sensitive data is restricted. In response, the authors propose and evaluate quantile regression attacks as a more effective alternative, demonstrating superior performance across image, tabular, and text domains. Theoretical analysis supports the generalization capability of quantile regression under distribution shifts caused by missing classes. Empirical results show significant gains in true positive rate (TPR) at low false positive rates (FPR), with up to 11× improvement over shadow models on CIFAR-100.

**Strengths:**

1. Identification of a Novel and Realistic MIA Setting: The motivation is grounded in real-world AI safety concerns, citing legal and ethical barriers to accessing harmful content for audit purposes.
2. Theoretical Justification for Generalization: The transferability theorem (Theorem 5.3) provides a formal condition under which quantile predictors trained on seen classes remain calibrated on unseen ones, linking performance to linear density ratios in feature space.

**Weaknesses:**

1. In Figure 3, the authors evaluate quantile regression against only the marginal baseline and random guessing. However, comparisons to shadow model–based baselines (e.g., LiRA or RMIA) under the same data-scarcity conditions  are absent. This omission weakens the empirical argument in Section 4.1.
2. While the paper identifies an important failure mode of existing MIAs, it does not sufficiently explore whether this issue is best understood as a domain shift problem—and if so, whether domain adaptation or calibration techniques could rescue shadow models. This paper is more likely to be a benchmark paper ranther than a methodology paper, so the absence of other baselines limits the paper’s contribution as a comprehensive benchmark.
3. Superficial empirical validation of theoretical conditions.
 - The paper uses low-dimensional (e.g., PCA or t-SNE) visualizations and fits a linear model to approximate the density ratio between seen and unseen class embeddings, reporting low MSE as evidence. However, in highly reduced dimensions, a linear (or even higher-order polynomial) fit may appear accurate even when the true high-dimensional relationship is nonlinear or the distributions are not genuinely aligned. Thus, observing a good linear fit in 2D does not substantiate the theoretical assumption that the density ratio is linear in the original feature space.
 - The condition may not justify quantile regression’s advantage: Even if the linear density ratio assumption holds, this would imply a structured, learnable relationship between seen and unseen classes—precisely the setting where standard domain adaptation or calibration techniques (e.g., recalibrating LiRA’s per-class thresholds using a small validation set or applying affine corrections) could mitigate shadow models’ failure. The paper does not compare against such adapted baselines, so the claimed necessity or superiority of quantile regression under this condition remains unsubstantiated.

**Questions:**

See weakness

---

> ### Author Response · Authors · 2025-11-21
>
> We thank the reviewer for their time and detailed feedback.
>
> >> **In Figure 3, the authors evaluate quantile regression against only the marginal baseline and random guessing. However, comparisons to shadow model–based baselines (e.g., LiRA or RMIA) under the same data-scarcity conditions are absent. This omission weakens the empirical argument in Section 4.1.**
>
> We would like to point out that the data scarcity experiment is on ImageNet, where shadow models are notoriously computationally expensive to train. Doing a full comparison against all of the data scarcity settings we demonstrate would require training hundreds of ImageNet classification models from scratch, which is not feasible given our computational restrictions. (See also the cited work Bertran et al. 2023, which shows that quantile regression attacks outperform LiRA in the fully-seen setting on ImageNet, but references the results of Carlini et al. 2022 as the comparison point due to computational limitations.)
> Nevertheless, we expect that our argument in Section 3 regarding the failures of shadow models would still hold in a setting where 90+% of classes are unseen.
>
> We would also like to point out that the single-class-drop setting is the most relevant for the CSAM detection setting that we focus on. While performance in the extreme data scarcity setting is an added benefit of quantile regression attacks, the main goal of our paper is to develop an attack to realistically address the CSAM setting, where the offending/unavailable data is in the minority (more similar to the CIFAR experiment settings where we do compare against shadow models and show that they cannot adequately address this setting).
>
> >> **While the paper identifies an important failure mode of existing MIAs, it does not sufficiently explore whether this issue is best understood as a domain shift problem—and if so, whether domain adaptation or calibration techniques could rescue shadow models. This paper is more likely to be a benchmark paper ranther than a methodology paper, so the absence of other baselines limits the paper’s contribution as a comprehensive benchmark.**
>
> While we are open to the possibility of using calibration techniques to adjust the shadow model-based thresholds, we do not directly see how they can apply. In our setting, we do not have access to unseen classes at attack training time, and thus, assuming access to even a small validation set is beyond our scope.
>
> If the reviewer has suggestions on specific domain adaptation or calibration techniques we could apply to improve the performance of shadow models in the fully-unseen-class setting, we are happy to test them and update our paper accordingly.
>
> >> **Superficial empirical validation of theoretical conditions.**
> >> **The paper uses low-dimensional (e.g., PCA or t-SNE) visualizations and fits a linear model to approximate the density ratio between seen and unseen class embeddings, reporting low MSE as evidence. [...] Thus, observing a good linear fit in 2D does not substantiate the theoretical assumption that the density ratio is linear in the original feature space.**
>
> We would first like to clarify a potential technical misunderstanding: the goal is not to identify a linear relationship between seen and unseen class embeddings, but rather between the set of seen class embeddings and all embeddings (seen + unseen).
>
> We would like to point out that the goal of the empirical validation is not to prove conclusively that a linear relationship exists, but rather to provide a best-effort estimate that corroborates our theoretical model given the sample sizes of the datasets and computational limitations. If the reviewer has suggestions on how to validate our hypotheses in higher dimensions in a computationally efficient way, given the limited sample sizes, we would be very happy to explore those as well.
>
> >> **The condition may not justify quantile regression’s advantage: Even if the linear density ratio assumption holds, this would imply a structured, learnable relationship between seen and unseen classes [...] The paper does not compare against such adapted baselines, so the claimed necessity or superiority of quantile regression under this condition remains unsubstantiated.**
>
> Again, we are unclear on how one would apply domain adaptation or calibration techniques in the unseen-class setting, where no validation set is present when training the attack. The key takeaway from our theory is that the quantile regression attack can perform well as long as a linear relationship exists, regardless of whether it can be observed or learned at training time.
>
> If our response clarifies our thought process and the reviewer's concerns, we would appreciate the reviewer revising their score accordingly. Thank you!

---

### Official Review · Reviewer_AJSX · 2025-10-31

**Soundness:** 2
**Presentation:** 3
**Contribution:** 2
**Rating:** 2
**Confidence:** 4

**Summary:**

The paper studies the problem of performing membership inference on data from unseen classes, where standard membership inference attacks based on training shadow models cannot have enough training data from the unseen classes. Under such setting, the authors found that quantile regression based MIA outperforms shadow model based counterparts, for TPR at small FPR metrics. Ablation experiments show that the proposed attack's power nicely interpolates under increasingly large number of unseen classes. Theoretical analysis for a linear quantile regression predictor shows that if the embedding does not change with and without the unseen classes, then the a learned calibrated quantile regression remains calibrated when applied to unseen classes, shedding light on the reason for the effectiveness of quantile regression MIA on unseen classes.

**Strengths:**

- MIA for unseen classes is a practical yet under explored threat model, e.g., in the context of CSAM detection. The proposed MIA method show promises of better generalization to unseen classes compared to shadow-model-based approaches, and is supported by analysis of linear quantile regression predictor under certain assumptions.
- Experiments cover a good range of Tabular, image and textual learning dataset.

**Weaknesses:**

- It is mainly a comparison and analysis paper, where all evaluated MIA methods are existent. The authors did not propose any new MIAs that could potentially boost MIA on unseen classes, but rather just applied quantile regression MIA and argued it is better than shadow-model-based method.
- Lack of comparison to shadow-model-free MIA baselines: this includes per-class population attack [Nasr et al. 2019, Ye et al. 2022] which is applicable when adversary has access to even a small pool of target class's data, Neighborhood attack (Mattern et al., 2023) and its similar variant for image (Choquette-Choo et al., 2021), and more text-based MIA methods including Min-K (Shi et al., 2023).
- The comparisons with shadow-model-based MIAs and RMIA make unrealistic design choices of not allowing MIAs to use ANY samples from unseen classes to train the reference models. In practice, it is often not the case and it is more realistic to assume adversary has access to a SMALL set of samples from target class. Although the authors argue due to legal reasons, training with even small number of samples from unseen classes may not be feasible. But to ensure fair and more realistic comparison, and for research purposes, the authors should also evaluate and report these MIAs when allowing one to include small set of samples from unseen classes in training shadow models.


References:
- Nasr, Milad, Reza Shokri, and Amir Houmansadr. "Comprehensive privacy analysis of deep learning: Passive and active white-box inference attacks against centralized and federated learning." 2019 IEEE symposium on security and privacy (SP). IEEE, 2019.
- Ye, J., Maddi, A., Murakonda, S. K., Bindschaedler, V., & Shokri, R. (2022, November). Enhanced membership inference attacks against machine learning models. In Proceedings of the 2022 ACM SIGSAC conference on computer and communications security (pp. 3093-3106).
- Mattern, J., Mireshghallah, F., Jin, Z., Schölkopf, B., Sachan, M., & Berg-Kirkpatrick, T. (2023). Membership inference attacks against language models via neighbourhood comparison. arXiv preprint arXiv:2305.18462.
- Shi, W., Ajith, A., Xia, M., Huang, Y., Liu, D., Blevins, T., ... & Zettlemoyer, L. (2023). Detecting pretraining data from large language models. arXiv preprint arXiv:2310.16789.
- Choquette-Choo, C. A., Tramer, F., Carlini, N., & Papernot, N. (2021, July). Label-only membership inference attacks. In International conference on machine learning (pp. 1964-1974). PMLR.

**Questions:**

See weaknesses.

---

> ### Author Response · Authors · 2025-11-21
>
> We thank the reviewer for their time and detailed feedback, and for recognizing the novelty of the threat model and the comprehensive evaluation.
>
> >> **It is mainly a comparison and analysis paper, where all evaluated MIA methods are existent. The authors did not propose any new MIAs that could potentially boost MIA on unseen classes, but rather just applied quantile regression MIA and argued it is better than shadow-model-based method.**
>
> We understand the reviewer's concern. However, we respectfully disagree that our work is as simple as "just applying quantile regression MIA." We are the first to systematically show any shortcoming of shadow model-based methods beyond the computational cost -- a contribution in itself. Beyond this, we then make a nontrivial observation that quantile regression MIA has an advantage in this setting and provide a theoretical analysis as to why this is the case.
> The proposed MIA setting itself is highly applicable in an important real world setting and has not been formalized previously.
>
> >> **Lack of comparison to shadow-model-free MIA baselines: this includes per-class population attack [Nasr et al. 2019, Ye et al. 2022] which is applicable when adversary has access to even a small pool of target class's data, Neighborhood attack (Mattern et al., 2023) and its similar variant for image (Choquette-Choo et al., 2021), and more text-based MIA methods including Min-K (Shi et al., 2023).**
>
> We appreciate the reviewer pointing out potentially related work. We carefully examined each of these papers, and below state our understanding of their relationship to our work. Based on our understanding, none of the cited methods apply in the setting we have proposed; we will include a discussion on these works in our revision to make this more clear.
>
> * Nasr et al. 2019.
>
> This is a white-box attack, which does not apply in our (black-box) setting.
>
> * Ye et al. 2022.
>
> The attacks in this paper are superseded/outperformed by the current state of the art high-power RMIA attack that we evaluate (see Zarifzadeh et al. 2024) and suffer the same weaknesses.
>
> * Mattern et al. 2023.
>
> This attack applies to generative text models, which are out of scope for our paper. We can see how a similar attack may be applicable in the unseen class setting for generative models, but a study of unseen class MIA for generative models (text or images) would merit a separate paper.
>
> * Choquette-Choo et al. 2021.
>
> This attack is not shadow-model-free; it requires training a shadow model on samples from the full data distribution (which would include the unseen class data).
>
> * Shi et al. 2023.
>
> This attack applies to generative text models, which are out of scope for our paper. Moreover, Min-k Prob has been brought into question as an effective attack by multiple works (Duan et al. 2024, Das et al. 2025).
>
> Das, Debeshee, Jie Zhang, and Florian Tramèr. "Blind baselines beat membership inference attacks for foundation models." IEEE Security and Privacy Workshops (SPW), 2025.
>
> Duan, Michael, et al. "Do membership inference attacks work on large language models?." Conference on Language Models (COLM), 2024.
>
> Zarifzadeh, Sajjad, Philippe Liu, and Reza Shokri. "Low-cost high-power membership inference attacks." International Conference on Machine Learning (ICML), 2024.
>
> Please see the next comment for a response to the remaining concern.

---

> > ### Author Response · Authors · 2025-11-21
> >
> > >> **The comparisons with shadow-model-based MIAs and RMIA make unrealistic design choices of not allowing MIAs to use ANY samples from unseen classes to train the reference models. In practice, it is often not the case and it is more realistic to assume adversary has access to a SMALL set of samples from target class. Although the authors argue due to legal reasons, training with even small number of samples from unseen classes may not be feasible. But to ensure fair and more realistic comparison, and for research purposes, the authors should also evaluate and report these MIAs when allowing one to include small set of samples from unseen classes in training shadow models.**
> >
> > We appreciate the reviewer's concern here. We proposed a specific problem setting based on discussions with collaborators in the field who corroborated that the *fully-unseen-class* setting is the most relevant and realistic in practice for the application of CSAM detection. We think our work is important in pulling out specific challenges in this new setting, demonstrating that the SoTA methods cannot be applied, and showing that the quantile regression attack does apply.
> >
> > As such, we do not see our work as being *unfair* to the baseline methods. We compare all methods on equal footing and in fact give unfair advantages to the baselines (such as knowledge of the target architecture).
> > In fact, we also give RMIA access to a limited set of unseen-class samples at evaluation time, which is exactly what the reviewer suggests (without deviating unrealistically far from our threat model). Even under this unfair advantage, RMIA does not perform as well as quantile regression.
> >
> > Nevertheless, we would like to understand better why the reviewer believes that the limited-sample-access setting is more realistic. Based on our understanding of the legal restrictions around CSAM, we have proposed the most relevant and realistic setting and we very much hope to inspire further research in this setting by focusing our paper narrowly on only this setting, rather than a broader study of the strengths and weaknesses of quantile regression attacks. If there is reason to believe our understanding of the legal aspects is incorrect we are open to discussion.
> >
> > If our response clarifies our thought process and the reviewer's concerns, we would appreciate the reviewer revising their score accordingly. Thank you!

---

### Official Review · Reviewer_5cCc · 2025-11-01

**Soundness:** 3
**Presentation:** 3
**Contribution:** 3
**Rating:** 6
**Confidence:** 4

**Summary:**

This paper introduces the "unseen class" setting for membership inference attacks. Here, an attacker predicts if examples from certain examples were used in training but the attacker has no access to samples from those classes when training the shadow models. Baseline schemes wildly fail, but this paper shows that by applying quantile regression attack (that learn features distinguishing members from non-members rather than modeling reference distributions) achieve up to 10x higher true positive rates prior methods.

**Strengths:**

The paper introduces a new threat model that's compelling. Unseen classes are a new interesting attack angle.

The paper introduces a 10x better method that does far better than prior techniques.

The ROC curves are useful to see, the results are convincing, the evaluation is well performed. There are no significant errors in anything.

**Weaknesses:**

This paper doesn't introduce anything really that new. The core method of quantile regression isn't anything that new. The overall scheme is basically applied exactly in the normal way. There's one small difference: instead of using the confidence on the true label (which requires knowing the ground truth label that may be from an unseen class), the paper uses the difference between the top two logits. But with that exception, the techniques are all the same. This isn't a terribly bad limitation, because it works. But it's not fully new.

I didn't find the introduction compelling for "the real-world AI safety scenario of detecting whether child sexual abuse material (CSAM) was used in a model’s training data." For this to work (1) the attacker needs to have come CSAM they want to test (which is illegal itself) but even more (2) wouldn't this have to assume that the original model actually has a "CSAM" class which seems .... unlikely? I don't understand this.

**Questions:**

Do you think you have anything that's technically novel in the methodology?

Can you come up with a compelling threat model where unseen classes are common?

---

> ### Author Response · Authors · 2025-11-21
>
> We thank the reviewer for their positive assessment and constructive feedback. Below, we address each main concern.
>
> >> **Do you think you have anything that's technically novel in the methodology?**
>
>
> We agree with the reviewer that the core method of quantile regression has been previously established. However, our work provides a number of other important (non-methodological) contributions: We motivate and formalize the unseen-class problem setting; show that the status quo methods fail in these settings (a particularly significant result given the general dominance of shadow-model-based methods); identify quantile regression as a promising approach and provide a grounded theoretical understanding of why quantile regression is robust to class dropout; and perform a thorough empirical analysis of this phenomenon across a variety of domains (text, image, tabular) and datasets. Building on this foundation, it may be interesting for future works to consider new variants of the quantile regression technique that are further robust to class dropout.
>
>
> >> **Can you come up with a compelling threat model where unseen classes are common?**
>
> Although our experiments are conducted on classification models, we are also generally interested in the ability to audit unseen classes across other model architectures, such as generative models. We focus on classification due to our desire to compare rigorously against shadow model baselines, which are prohibitively expensive to train for generative models in the large number of settings we study. While having a "CSAM class" may sound unrealistic in a classification setting, this is a clean, abstracted version of the generative setting, in which one can indeed label a training image or a generated image to be CSAM or not CSAM. Similar confidence-based approaches can also be effective in this setting [1], and we anticipate that our findings will be applicable, but
>
> In the setting we explore, a third-party developer or model auditor seeks to develop a tool/attack that law enforcement agencies can use to audit models. The attacker would not legally be able to access the CSAM they want to test, and would instead be creating an MIA for trusted authorities who do have access (to a small pool of query samples). This is a realistic setting as the organizations (i.e., law enforcement agencies) responsible for data and model auditing may not have the technical expertise to design and train an MIA themselves; this expertise instead frequently comes from external partnerships.
>
> We hope that these clarifications resolve the remaining concerns about novelty and motivation, and we believe they support a higher overall assessment of the paper.
>
> [1] Tang, Shuai, et al. "Membership Inference Attacks on Diffusion Models via Quantile Regression." International Conference on Machine Learning. PMLR, 2024.

---

### Official Review · Reviewer_Y5XR · 2025-11-01

**Soundness:** 3
**Presentation:** 3
**Contribution:** 3
**Rating:** 4
**Confidence:** 3

**Summary:**

This paper studies membership inference attacks in the "unseen class" setting, where attackers lack access to certain classes during training but must infer membership on those classes. This scenario captures real-world AI safety auditing constraints (e.g., CSAM detection, medical record auditing). The authors demonstrate that shadow model-based attacks fail catastrophically in this setting, while quantile regression attacks achieve superior performance by learning features that generalize across classes. Theoretical analysis based on multi-accuracy explains this generalization, supported by empirical validation.

**Strengths:**

-  Important Problem: Identifies a critical yet unstudied scenario in practical AI safety auditing with significant real-world implications.
-  Systematic Evaluation: Comprehensive experiments across image, text, and tabular datasets demonstrate quantile regression's consistent superiority.
-  Theoretical Support: Provides theoretical explanation for why quantile regression generalizes to unseen classes.

**Weaknesses:**

-  Insufficient Failure Analysis: Improvements are minimal on some datasets (e.g., CINIC-10). The paper lacks analysis of when quantile regression fails or how much data diversity ensures generalization.
-  Unrealistic Assumptions: Assumes complete knowledge of target model architecture and training process. No systematic evaluation of robustness to architecture mismatch or training differences. RMIA receives unfair advantage (using unseen class samples at evaluation) without adequate discussion of fairness impact.

**Questions:**

This paper makes a valuable contribution by revealing shadow models' failure in the unseen class setting and proposing quantile regression as an effective alternative. However, improvements are needed: (1) deeper analysis of failure conditions and quantitative relationships between dataset characteristics and attack performance, (2) systematic robustness evaluation under realistic "black-box" auditing conditions with architecture/training mismatches, and (3) fairer experimental comparisons by addressing RMIA's evaluation advantage. Strengthening these aspects will significantly enhance the paper's depth and practical value.

---

> ### Author Response · Authors · 2025-11-21
>
> We thank the reviewer for their constructive feedback, and for recognizing the importance of the problem, comprehensive evaluation, and theoretical model. Below, we address each main concern.
>
> >> **Insufficient Failure Analysis: Improvements are minimal on some datasets (e.g., CINIC-10). The paper lacks analysis of when quantile regression fails or how much data diversity ensures generalization.**
>
> We appreciate the reviewer's concerns.
>
> However, we respectfully disagree that we did not provide a failure analysis for quantile regression. For the CINIC-10 dataset, we have provided an explanation for the minimal performance gain in Section 5. We point the reviewer to Figure 5c, where we relate the performance on CINIC-10 to our theoretical model. The PCA projection of the feature embeddings indicates that there is *not* a linear relationship between the seen class set and the (seen+unseen) class set. This provides evidence that we would not expect quantile regression to perform well in the CINIC-10 setting, as we observe empirically. In contrast, in Figures 5a and 5b, we see that the unseen classes in our CIFAR-100 and ImageNet experiments have a stronger linear relationship with the seen classes, corroborating the attack's stronger performance in those settings.
>
> We are happy to discuss further if we have misunderstood the reviewer's points here, but we believe our model and analysis in Section 5.1 address these concerns.
>
> >> **Unrealistic Assumptions: Assumes complete knowledge of target model architecture and training process. No systematic evaluation of robustness to architecture mismatch or training differences. RMIA receives unfair advantage (using unseen class samples at evaluation) without adequate discussion of fairness impact.**
>
> We would first like to clarify a potential *misunderstanding* here. In fact, quantile regression MIA is a black-box method that *does not* have knowledge of the target model architecture nor training process.
>
> From there, we followed standard evaluation best practice, which is to give the baseline methods the *most advantage* and the proposed alternative the *least advantage* in evaluation in order to give the *most fair* comparison. That is, the baseline methods (LiRA, RMIA) are given knowledge of the target model architecture as well as access to unseen classes at evaluation time to give them the *most advantage*, yet they are *still outperformed* by the quantile regression attack, which has access to neither.
>
> Removing these advantages would only degrade the performance of LiRA and RMIA and not change the performance of the quantile regression attack, therefore not changing our conclusions. Nevertheless, we are happy to add experiments in these settings if the reviewer still thinks they would add value to our findings.
>
> If our comments clarify the reviewer's concerns, we would very much appreciate if the reviewer could revise their score accordingly.

---

### Meta-Review · Area_Chair_7Kpr · 2026-01-07

**Summary:**

The paper introduces a new “unseen‑class” threat model for membership inference attacks, where the attacker has no data from certain target classes, and demonstrates that traditional shadow‑model attacks break down in this setting; it proposes a quantile‑regression attack that relies on a simple top‑two‑logit‑difference feature and a single lightweight regressor, achieving markedly higher true‑positive rates at low false‑positive rates across image, tabular, and text datasets compared to shadow models and a basic global‑threshold baseline; the authors support the method with a theoretical result showing that linear density‑ratio relationships between seen and all classes guarantee calibration transfer, and provide empirical evidence of such linearity for most evaluated datasets; reviewers highlight the relevance of the problem, the cross‑domain empirical validation, the theoretical insight, and the computational efficiency, while noting limited analysis on failure cases, assumptions about attacker knowledge, modest novelty of the regression technique, and the absence of certain baseline comparisons.

**Reviewer Concerns:**

- All reviewers acknowledge that the paper tackles a new “unseen‑class” threat model that is relevant to high‑stakes privacy auditing.
Reviewers Y5XR, 5cCc, AJSX and C19G each raise concerns about the realism of the threat model (e.g., the CSAM scenario) and note that the baseline MIAs receive an unfair advantage yet still underperform.

- Reviewers AJSX and C19G both point out the lack of comparison to other shadow‑model‑free baselines (per‑class population, Neighborhood, Min‑K) and the missing shadow‑model experiments under data‑scarcity.

- Reviewers AJSX (rating 2) and C19G (rating 2) consider the paper insufficient, citing lack of methodological novelty, unrealistic threat model, and missing baseline/comparison experiments, and they recommend rejection.

**Reviewer Scores:**

- To the main concerns of AJSX (rating 2) and C19G (rating 2, the authors replied that (1) per‑class‑population, Neighborhood and Min‑K attacks are not applicable because they require white‑box access or unseen‑class samples, which are unavailable in the proposed threat model, and they emphasized that even when baselines are given “unfair” advantages (knowledge of the target architecture and access to unseen‑class samples at evaluation) they still underperform; and (2) training hundreds of ImageNet shadow models is computationally infeasible, so they cited prior work (Bertran et al., 2023) showing quantile‑regression already outperforms LiRA on ImageNet in the fully‑seen setting, argued that the main contribution is exposing the shadow‑model failure mode (domain‑adaptation or calibration would require a validation set from unseen classes, contradicting the threat model), and clarified that their linear‑density‑ratio analysis is performed between seen embeddings and the combined seen + unseen embeddings, with PCA visualisations provided as the best‑effort estimate given limited samples.

- I am not confident the answer would have changed the reviewers' score.

---

### Decision · Program_Chairs · 2026-01-26

Reject